# Signaling Pathways Involved in Manganese-Induced Neurotoxicity

**DOI:** 10.3390/cells12242842

**Published:** 2023-12-14

**Authors:** Hong Cheng, Beatriz Ferrer Villahoz, Romina Deza Ponzio, Michael Aschner, Pan Chen

**Affiliations:** Department of Molecular Pharmacology, Albert Einstein College of Medicine, Bronx, NY 10461, USA; hong.cheng@einsteinmed.edu (H.C.); beatriz.ferrervillahoz@einsteinmed.edu (B.F.V.); romina.dezaponzio@einsteinmed.edu (R.D.P.); michael.aschner@einsteinmed.edu (M.A.)

**Keywords:** manganese, neurotoxicity, signaling pathway, insulin-like growth factor (IGF), autophagy

## Abstract

Manganese (Mn) is an essential trace element, but insufficient or excessive bodily amounts can induce neurotoxicity. Mn can directly increase neuronal insulin and activate insulin-like growth factor (IGF) receptors. As an important cofactor, Mn regulates signaling pathways involved in various enzymes. The IGF signaling pathway plays a protective role in the neurotoxicity of Mn, reducing apoptosis in neurons and motor deficits by regulating its downstream protein kinase B (Akt), mitogen-activated protein kinase (MAPK), and mammalian target of rapamycin (mTOR). In recent years, some new mechanisms related to neuroinflammation have been shown to also play an important role in Mn-induced neurotoxicity. For example, DNA-sensing receptor cyclic GMP–AMP synthase (cCAS) and its downstream signal efficient interferon gene stimulator (STING), NOD-like receptor family pyrin domain containing 3(NLRP3)-pro-caspase1, cleaves to the active form capase1 (CASP1), nuclear factor κB (NF-κB), sirtuin (SIRT), and Janus kinase (JAK) and signal transducers and activators of the transcription (STAT) signaling pathway. Moreover, autophagy, as an important downstream protein degradation pathway, determines the fate of neurons and is regulated by these upstream signals. Interestingly, the role of autophagy in Mn-induced neurotoxicity is bidirectional. This review summarizes the molecular signaling pathways of Mn-induced neurotoxicity, providing insight for further understanding of the mechanisms of Mn.

## 1. Introduction

Despite concerted efforts to reduce heavy metal pollution over the past few decades, exposure to heavy metals still poses a significant threat to public health [1]. Due to natural activities or anthropogenic industrial activities, large amounts of heavy metals have been discharged into the environment. Humans are exposed to heavy metals through food, water, and air [2,3]. Heavy metals accumulate in multiple target organs, and even exposure to low levels may cause health effects [4].

Manganese (Mn) is an essential trace element involved in the synthesis and activation of many enzymes, including glutamine synthetase, pyruvate decarboxylase, manganese superoxide dismutase (MnSOD), and arginase [5]. Mn is involved in the regulation of amino acid, lipid, and carbohydrate metabolism [6]. Mn enters the human body mainly through the respiratory and digestive tracts. Under normal circumstances, an adult’s daily intake of Mn is in the order of 2–6 mg, of which 1–5% is absorbed by the duodenum. It enters the enterohepatic circulation through the portal vein and then enters the gastrointestinal tract via biliary secretions, and it is finally excreted in the feces [7].

Mn deficiency and intoxication are both associated with adverse neuropsychiatric effects, but Mn deficiency is extremely rare [8]. Excessive exposure to Mn is mainly related to occupational exposures, including welding, mining, smelters, paints, dry batteries, and Mn-rich agricultural chemicals [9,10,11]. The high concentrations of Mn in the environment are due to its abundance in the Earth’s crust, the release of Mn-containing exhaust gases and wastewater into the environment by anthropogenic activities, and the use of gasoline additives (methylcyclopentadienyl manganese tricarbonyl (MMT)) and fungicides [12,13,14,15]. Infant Mn exposure is mainly associated with excess Mn supplementation in parenteral nutrition [16].

The brain is the most sensitive target organ for Mn toxicity. Mn exists in 11 oxidation states from −3 to +7, among which Mn^2+^ and Mn^3+^ are commonly found in biological systems [17]. Mn enters the brain through the blood–brain barrier (BBB) and blood–cerebrospinal fluid barrier (BSCFB), primarily in the form of Mn^2+^, Mn-citrate, Mn^3+^-transferrin, or alpha-2- macroglobulin [18]. Mn mainly accumulates in the basal ganglia (striatum, pallidum, and substantia nigra), hippocampus, thalamus, and cortex [19]. Excessive exposure to Mn can lead to manganism, which is characterized by symptoms similar to Parkinson’s disease (PD), including gait disturbance, dystonia, and postural tremor [20]. In addition, Mn exposure has been associated with cognitive deficits, attention loss, and neuropsychological abnormalities [21]. Moreover, excessive exposure to Mn early in life can lead to learning disabilities, hyperactivity, and attention deficit disorders, and may increase the risk of neurodegenerative diseases later in life [22,23,24,25].

In addition to manganism, multiple studies demonstrated a significant association between Mn and neurodegenerative diseases. Elevated Mn levels were observed in patients with PD [26] and amyotrophic lateral sclerosis (ALS) [27], but Mn deficiency has been observed in Huntington’s disease (HD) [28], and the relationship between Mn levels and Alzheimer’s disease (AD) remains controversial [29]. However, due to the excretion of Mn, the time point and biomarkers of assessment may affect the results of these studies.

Due to the significant effects of Mn exposure on the nervous system, the mechanisms of Mn-induced neurotoxicity have been extensively studied, including neuroinflammation, protein homeostasis [30], mitochondrial function and REDOX homeostasis [31], calcium homeostasis [32], neurotransmitter metabolism [33], microRNA (miRNA) function [34], and metal homeostasis [35]. At present, the understanding of the underlying mechanisms of Mn neurotoxicity is rapidly developing. Several studies revealed the insulin-mimicking effects of Mn, which can activate several of the same metabolic kinases and directly increase peripheral and neuronal insulin and insulin-like growth factor (IGF)-1 levels in rodent models [36,37,38,39]. As an important cofactor of many kinases and phosphatases, Mn plays a key role in cell signaling pathways. Mn can activate mitogen-activated protein kinase (MAPK), protein kinase B (PKB/Akt), ataxia telangiectasia mutated (ATM), and mammalian target of rapamycin (mTOR) both in vivo and in vitro. Since these kinases regulate transcription factors, such as cyclic adenosine monophosphate (cAMP), cAMP response element-binding protein (CREB), p53, nuclear factor κB (NF-κB), and forkhead box O (FOXO), Mn can also regulate cellular function at the transcriptional level [40,41,42]. Therefore, it is important to study the role of Mn homeostasis and related signaling pathway in Mn essentiality and toxicity. However, it is not fully understood which Mn-dependent enzymes are most sensitive to changes in Mn homeostasis and the relationship between Mn and these signaling pathways. Therefore, this review focuses on cell signaling pathways associated with Mn neurotoxicity, including the classical insulin/IGF signaling pathway, neuroinflammatory pathways, and autophagy, which is an important downstream protein degradation pathway.

## 2. Insulin and Insulin-like Growth Factor (IGF) Signaling Pathway

Insulin and IGF are homologous growth hormones that regulate cell metabolism, and their main function is to increase anabolism and reduce catabolism. Insulin has short-term metabolic effects, such as increasing glucose and amino acid transport, inducing glycolysis, glycogenesis, lipogenesis, and protein synthesis, and inhibiting gluconeogenesis, lipolysis, and protein degradation. However, IGF has chronic effects that determine cell fate, such as inducing proliferation, inhibiting apoptosis, and inducing differentiation [43]. In the brain, insulin and IGF are necessary for synaptic maintenance and activity, neurogenesis, neurite growth, and mitochondrial function [28]. Therefore, the dysregulation of these neurotrophic factors is believed to be associated with neurodegenerative diseases, and abnormal IGF/ insulin levels and insulin-related signaling changes have been observed in neurodegenerative diseases, including PD [44], AD [45], HD [28], and ALS [46].

Upon the binding of insulin or IGF receptor (IR/IGFR) on the target cell, the receptor’s tyrosine kinase is activated to phosphorylate several specific substrates, particularly insulin receptor substrates (IRSs) and Src homologous collagen (SHC) [47]. Phosphorylated tyrosine residues of these substrates are recognized by various signaling molecules containing the Src Homology 2 (SH2) domain. For example, 85 kDa regulatory subunits (p85) of phosphatidylinositol 3-kinase (PI3K), growth factor receptor binding 2 (GRB2), and protein tyrosine phosphatase 2 containing sh2 (SHP2/Syp) [48]. These bindings activate downstream signaling pathways, the PI3K pathway, and the Ras mitogen-activated protein kinase (MAPK) pathway [49]. Activation of these well-known signaling pathways is required to induce the various biological activities of IGF [50].

Mn has been shown to activate several of the same pathways as insulin/IGF, including AKT, MAPK, and mTOR, and even the IR/IGFR itself [28]. Several in vivo studies found that Mn deficiency caused glucose intolerance and reduced insulin secretion, manifested by decreased circulating IGF-1 and insulin, and increased IGFBP3 [36,37,38,39,51]. Mn supplementation improves glucose intolerance and prevents diet-induced diabetes by increasing insulin secretion and the expression of IGFR and IGF-1 in the hypothalamus, as well as increasing the expression of MnSOD [52,53,54]. These results are consistent with reports of diabetic patients responding to oral Mn therapy, as well as reduced blood Mn levels in diabetic patients [55,56].

However, few studies focused on the role of the IGF receptor itself in Mn toxicity. Tong and his colleagues found that Mn exposure reduced the expression of ATP and insulin/IGF receptors [57]. In addition, Srivastava and his colleagues found that Mn stimulated hypothalamic IGF-1 dose–dependent release and increased p-IGF-1R expression in vitro and affected hypothalamic development in adolescent rats through IGF-1/Akt/mTOR pathways [54]. Other studies demonstrated that Mn could regulate two important downstream signaling pathways of the IGF, PI3K/Akt, and MAPK signaling pathways.

### 2.1. PI3K/Akt Signaling Pathway

PI3K/Akt can be activated by a variety of growth factors and plays a central role in cell growth regulation, proliferation, metabolism, and cell survival, as well as neuroplasticity [58]. The interaction between the p85 and IGF-IR substrate leads to the activation of PI3K. Activated PI3K phosphorylates PIP2 to produce PIP3. PIP3, in turn, activates PDK1. PDK1 phosphorylates Akt threonine 308 residues and mTORC2 phosphorylates Akt serine 473 residues, resulting in complete Akt/PKB activation. Activated Akt phosphorylates a variety of Akt substrates, including AS160, Bad, forkhead box-containing protein, O subfamily (FoxO1), Tsc1, and glycogen synthase kinase 3β (GSK3β) [43].

PI3K inhibition reduces Mn uptake in mouse striatal cell lines (STHdh) [59]. In a Huntington’s disease cell model, exposure to both physiological (1nM) and over-physiological (10 nM) Mn levels enhanced p-IGFR/IR-dependent AKT phosphorylation, and more than 70% of Mn-induced p-Akt signaling was dependent on p-IGFR rather than other upstream and downstream effectors [60]. Our previous studies have shown that exposure to toxic doses of Mn can increase the phosphorylation levels of Akt in rat hippocampal tissue [61] and PC12 cells [62]. Moreover, pretreatment of PC12 cells with LY294002, a PI3K/Akt inhibitor, further increased apoptosis, suggesting that the Mn-activated PI3K/Akt signaling pathway played a protective role. Bae et al. [63] reported that the phosphorylation level of Akt in BV2 microglia increased after 500 μM Mn treatment for 1 h. Short-term (postnatal day (PND) 8–12) exposure to Mn increased Akt phosphorylation levels in rat striatum [23]. Mn exposure also increased Akt expression levels in *C. elegans* [64]. However, another study reported that Mn exposure inhibited the PI3K/Akt signaling pathway in cortical neurons, and the apoptosis induced by Mn can be improved by increasing the level of the PI3K/Akt signaling pathway [65]. Brain-derived neurotrophic factor (BDNF) is a key nutrient factor involved in neurobiological mechanisms of learning and memory, and both population studies and animal experiments have confirmed that Mn exposure can reduce BDNF levels [66,67]. Mn exposure can increase α-Synuclein (α-Syn) expression in mice, and the Mn-dependent enhancement of α-Syn expression can further exacerbate the decrease in BDNF protein level and inhibit TrkB/Akt/Fyn signaling, and thus interfere with FYN-mediated phosphorylation of NMDA receptor GluN2B subtyrosine [68]. Whether Mn exposure activates or inhibits the PI3K/Akt signaling pathway may be due to different doses and/or timing of exposure and the use of different animals or cell lines, but in general, activation of the PI3K/Akt signaling pathway protects against Mn-induced neurotoxicity.

Transcription factor FOXO family proteins, substrates of Akt, can be phosphorylated by active Akt and excluded from the nucleus [69,70], thereby inhibiting the transcription of genes associated with oxidative stress protection; for example, MnSOD-2, catalase, and anti-apoptotic effects (Bim and Fas ligands) [71,72,73,74,75]. Mn exposure can increase FOXO3a levels [62,76]. In *C. elegans*, resistance in AKT-1/2-deficient mutant strains to Mn toxicity was higher than in wild type N2 strains, possibly because the loss of AKT-1/2 reduced the inhibition of DAF-16, and thus increased the antioxidant response [77]. In contrast, another study [40] found that treating astrocytes with 100 or 500 μM of Mn resulted in increased FOXO (dephosphorylation and phosphorylation) levels, and p-FoxO disappeared from the cytoplasm after Mn exposure, while AKT phosphorylation levels remained unchanged, but MAPK phosphorylation levels, especially p38 and ERK, and PPARγ coactivator 1 (PGC-1) levels were elevated, suggesting that FOXO may be regulated by MAPK rather than the Akt signaling pathway.

### 2.2. MAPK Signaling Pathway

Protein kinases ERK1/2, JNK1/2, and p38MAPK are the most important enzymes of the MAPK family [78,79]. ERK1/2 is primarily activated by growth factors and regulates gene expression, embryogenesis, proliferation, cell death/survival, and neuroplasticity [80,81]. JNK1/2/3 and p38MAPK protein kinases are commonly known as stress-activated protein kinases (SAPKs), which can be activated by cytokines and cytotoxic damage and are associated with stress and cell death [82,83]. GRB2 interacts with the IGF1 receptor substrate to activate GRB2-associated SOS guanine nucleotide exchange activity, thereby activating Ras small GTPase, which, in turn, activates MAPK. This kinase cascade is called the Ras-MAPK pathway. Activated MAPK phosphorylates transcriptional activators, leading to the induction of various IGF biological activities [43].

Acute exposure to Mn (3–6 h) at concentrations that did not affect cell viability activated MAPKs (ERK1/2 and JNK1/2) in the hippocampus and striatum sections of immature rats (PND14) [84]. Mn (500 μM) exposure increased iNOS expression in BV2 microglia, and the increase in iNOS protein expression was mediated by the JNK-ERK MAPK and PI3K/Akt signaling pathways, but not by the p38 MAPK signaling pathway [63]. Sub-acute (4 weeks) and sub-chronic (8 weeks) exposure to Mn (15 mg/kg) resulted in an increased expression of p38, p-ERK, and p-JNK in the thalamus of rats. Treatment with sodium para-aminosalicylic acid (PAS-Na), an anti-inflammatory drug, decreased p-JNK and p-P38 levels but did not decrease p-ERK levels [85]. Mn exposure also enhanced the activation of the p38 MAPK and JNK pathways in PC12 cells and mouse brain tissue, but only the activation of p38 MAPK reduced Mn-induced neuronal apoptosis through BDNF regulation [86]. Short-term exposure to Mn (PND8-12) increased the phosphorylation of DARPP-32-Thr-34, ERK1/2, and AKT in the rat striatum. TroloxTM, an antioxidant, reversed the increase in ROS and ERK1/2 phosphorylation but failed to reverse the increase in AKT phosphorylation and motor deficit induced by Mn (20 mg/kg) [23], indicating that MAPK activation is related to oxidative stress but Akt activation is not directly related to oxidative stress.

cAMP is the second intracellular messenger. CREB is a target of CAMP-dependent protein kinase A (PKA), activated by the phosphorylation of PKA at Serine-133 (Ser133), and is also regulated by Ca^2+^ and p38 MAPK [87,88,89,90,91]. BDNF is downstream of the cAMP-PKA-CREB signaling pathway [92]. Mn exposure increased CREB and p38 MAPK phosphorylation and apoptosis in PC12 cells, while CREB knockdown decreased BDNF levels and increased Mn-induced apoptosis, suggesting that CREB activation had a protective effect on neuronal apoptosis. In addition, the inhibition of Ca^2+^ and p38 MAPK significantly reduced CREB phosphorylation levels [86]. Mn exposure may also reduce the level of BDNF by inhibiting the CAMP-PKA-CREB signaling pathway in hippocampal and PC12 cells, inducing apoptosis and leading to cognitive impairment [66,93]. Phosphoprotein DARPP-32 is highly expressed in spinous neurons in the striatum. The altered phosphorylation status of Thr-34 or Thr-75 in DARPP-32 gives it the unique properties of dual function as both an inhibitor of protein phosphatase 1 (PP1) and an inhibitor of PKA. Rats exposed to 5 or 10 mg Mn/kg on PND8-12 showed increased phosphorylation of DARPP-32 at Thr-34 in the striatum on PND14 [23].

## 3. Neuroinflammatory Signaling Pathway

### 3.1. cCAS-STING Signaling Pathway

Mn has emerged as an activator of the cyclic GMP-AMP synthase (cGAS) stimulator of interferon genes (STINGs) pathway. This pathway is an inductor of innate immune defense programs [94]. cGAS acts as a cytosolic double-stranded DNA (dsDNA) sensor. cGAS recognizes dsDNA and activates the STING through second messenger cyclic GMP-AMP (cGAMP) production [95]. Following activation, the STING translocates from the endoplasmic reticulum to the Golgi. Then, it recruits and activates TANK-binding kinase 1 (TBK1), which phosphorylates the transcription factor interferon regulatory factor 3 (IRF3) [96]. Following phosphorylation, IRF3 translocates to the nucleus, triggering the production of type I interferon (IFN) [94]. In addition, the STING also activates STAT6 [97] and NF-κB [98].

Cytosolic Mn enhances the cGAS-STING pathway, mediating host innate immune responses against DNA viruses [99]. Mn is released from damaged organelles (mitochondria and Golgi apparatus), accumulating in the cytosol, where it enhances the sensitivity of cGAS to dsDNA and the production of type I IFN [99]. Thus, Mn-insufficient mice showed heightened susceptibility to viral infection. Similarly, Mn-deficient mice-derived macrophages display higher vulnerability to DNA virus infection. Mn-treated cells partially restored the response to DNA virus [99]. In addition to dsDNA, Mn acts as a second cGAS activator without cytosolic DNA, inducing type I IFN production without viral infection [100,101]. In an in vivo tumor model, Mn administration reduced tumor burden, whereas cGAS knock-out mice and STING-deficient mice did not respond to Mn [102]. However, excessive Mn exposure induces neurotoxicity and alters cognitive and motor functions [103,104]. The cGAS-STING pathway emerged as a promising target to better understand the Mn neurotoxic mechanisms. In the CNS, microglial cells are the principal source of cGAS–STING-induced IFN [105]. In the BV-2 microglial cell line, Mn exposure (100 µM, 12 h) activated the cGAS-STING pathway. Interestingly, the STING selective inhibitor H151 treatment reduced Mn-induced microglial activation and IFN production, suggesting that the cGAS-STING pathway has a role in Mn-induced neurotoxicity [106].

A recent study showed that the STING binds the NOD-like receptor family pyrin domain containing 3 (NLRP3) and promotes inflammasome activation [107]. Mn induces host response through cGAS-STING and NLRP3 activation [108]. Thus, the cGAS-STING-mediated NLRP3 inflammasome has emerged as a new mechanism involved in Mn-induced neurotoxicity.

### 3.2. NLRP3- CASP1 Signaling Pathway

One of the Mn neurotoxic mechanisms is the induction of neuroinflammation [109,110]. Mn induces a proinflammatory state in CNS through inflammasome activation and Il1β release [110,111]. The NLRP3 inflammasome complex contains NLRP3 scaffold protein, apoptosis-associated speck-like protein (ASC), and pro-caspase1. NLRP3 oligomerizes and recruits ASC, leading to pro-caspase1 cleaving to the active form capase1 (CASP1) in an autocatalytic process. Next, active caspase1 mediates pro-IL1β and pro-IL18 protein hydrolysis to their matured forms (IL1β and IL18, respectively) [112]. NLRP3-CASP1 inflammasome formation needs two signals for its activation. The first signal (Signal 1) acts as a priming signal and involves NF-κB activation to induce NRPL3, pro-IL1β, and pro-IL18 expression. The second signal (Signal 2) triggers the assembly of NLRP3-CASP1 inflammasome and its activation. This second signal may implicate lysosomal dysfunction, mitochondrial damage, potassium leakage, and de-ubiquitination [113,114,115]. Recent works suggest that Mn acts as a signal 2 to activate the NRLP3-CASP1 inflammasome in microglial cells [111,116].

In vivo and in vitro studies demonstrated that Mn induces the NLRP3-CASP1 inflammasome pathway, triggering an innate immune response. Mn induces the NRLP3-CASP1 inflammasome pathway in the hippocampus of mice administered 100 mg/kg Mn subcutaneously three times for 1 week [116]. In male Sprague–Dawley (SD) rats, exposure to 2, 5, and 10 mg/kg Mn via daily gavage administration for 30 days increased NRLP3, ASC, and cleaved CASP1/CASP1 protein expression in the striatum [117]. Low-dose chronic exposures of Mn (1 and 5 mg/kg), with long recovery periods between administration (every 10 days) for 150 days, increased NRLP3 gene expression in the substantia nigra pars compacta (SNpc)-enriched midbrain in male SD rats [118]. Mn administration via drinking water (200 mg/L for 5 weeks) increased NLRP3 and cleaved-CASP1 and IL1β levels [119], supporting the previous studies.

Similarly, in the BV2 microglial cell line, 100 µM Mn treatment for 6 h activates NLRP3-CASP1 inflammasome. This activation releases lysosomal cathepsin B, triggering autophagy–lysosomal dysfunction [116]. Cathepsin B must play a role in Mn-induced NLPR3-CASP1 inflammasome. Mn also induces mitochondrial dysfunction, leading to NLRP3-CASP1 inflammasome activation. In this alternative mechanism, Mn (100 µM, 24 h) produces mitochondrial dysfunction in primary microglial cells primed with LPS for 3 h, decreasing the expression of the mitochondrial fusion protein 2 (mitofusin 2) and vacuolar protein sorting associate protein 35 (VPS35), a retromer complex protein that induces mitofusin 2 ubiquitination [111]. A recent study demonstrated a new mechanism that mediates NLRP3 inflammasome activation, where LRRK2 (leucine-rich repeat kinase 2) triggers Mn-induced NLRP3-CASP1 inflammasome activation in BV2 cells (250 µM for 6, 12, or 24 h) and male C57BL/6 mice (30 mg/kg Mn, 3 weeks) [120]. Moreover, the Mn-induced NLRP3-CASP1 inflammasome pathway in mouse neuroblastoma (N2a) cells (250, 500, and 1000 µM Mn for 24 h and 48 h) and Mn-exposed male SD rats (25 mg/kg Mn, 30 days) mediates KH-type splicing regulatory protein (KHSRP) expression [121].

PAS-Na exhibits anti-inflammatory properties and prevents the Mn-induced expression of NRLP3, CASP1, IL18, and IL1β in BV2 microglial cells treated with 200 µM Mn for 12 h prior to PAS-Na treatment [122]. Mn treatment (100, 200, and 400 µM for 24 h) also activated the NF-κB pathway in BV2 cells, whereas PAS-NA antagonized Mn effects [123]. In vivo experiments support this observation, with male rats exposed to Mn (5, 10, and 20 mg/kg, 14 weeks) showing activated NRLP3-CASP1 inflammasome and NF-κB pathway in the hippocampus and basal ganglia [123]. Experiments in highly aggressively proliferating immortalized (HAPI) rat microglial cells demonstrated that Mn (0, 100, 200, 300, and 500 µM for 12 h) induces NF-κB and NLRP3-CASP1 inflammasome. Pretreatment with NF-κB inhibitors reverses Mn-induced NLPR3 activation [117], suggesting that NF-κB mediated Mn-induced NRLP3-CASP1 inflammasome activation.

### 3.3. NF-κB Signaling Pathway

The NF-κB signaling pathway regulates cytokine and chemokine expression, orchestrating the inflammatory responses in microglia [124] and astrocytes [125]. NF-κB is a dimeric transcription factor formed by two of five subunits (p50, RelA/p65, c-Rel, RelB, and p65). The most common dimeric forms are p50 and p65. In resting conditions, NF-κB dimers are in the cytoplasm, sequestered by I kappa B kinase (IKK) [126]. Upon its activation, NF-κB translocates to the nucleus and regulates the expression of genes implicated in inflammation, cell growth, and cell death [126,127].

Mn potentiates nitric oxide synthase 2 (NOS2) activity through NF-κB activation in astrocytes [128] and microglia [124]. In glioma C6 cells, 300 µM Mn treatment for 24 h activates the NF-κB pathway and enhances NOS2 expression [128]. Moreover, Mn enhances lipopolysaccharide (LPS)-induced inflammatory response, NOS2 expression, and mitochondrial ROS production in glioma C6. Likewise, LPS-activated NOS2 is potentiated by Mn (up to 1000 µM, for 48 h)-induced NF-κB in N9 microglial cell line [124]. Inhibiting mitochondrial ROS production ameliorates Mn response in glioma C6 cells [128]. Similarly, a low Mn concentration (10 µM, for 8 h) potentiates NOS induction in primary cortical murine astrocytes primed by a mild inflammatory insult [125]. Thus, Mn increases LPS-activated NOS2 expression, inducing NF-κB activation and mitochondrial dysfunction. These studies suggest that Mn enhances inflammatory response via an NF-κB-dependent mechanism in glial cells. In vivo experiments corroborate these findings. In mice, Mn juvenile exposure (100 mg/kg, PND 21–34) increases NF-κB activation and NOS2 expression in basal ganglia [42]. Interestingly, female mice are more resistant to Mn juvenile exposure than males. However, supplementation with 17-β-estradiol (E2) prevents Mn-induced NF-κB activation and NOS2 expression in males. E2 protects against Mn-induced inflammation in developing mice and represents a mechanism that mediates the sex vulnerability to Mn neurotoxicity observed in children.

A recent study demonstrated a novel mechanism that mediates Mn-induced NF-κB activation. In C57BL/6 male mice (100 mg/kg Mn, subcutaneously injected three times in 1 week) and the BV2 microglial cell line (100 µM Mn for 12 h), Mn induced PSMB8 (an immunoproteasome catalytic subunit) expression and activity. PSMB8 inhibition decreased NF-κB p65 phosphorylation [129].

Mn dysregulates EAAT2 (excitatory amino acid transporter 2). The NF-κB signaling pathway mediates, in part, the alteration in EAAT2. In the H4 human astrocyte cell line, Mn exposure (250 µM for 3h) induces the phosphorylation of IKK2, NF-κB p65 translocation to the nucleus, and Ying Yang 1 (YY1) transcription, repressing EAAT2 [130].

The crosstalk between microglia and astrocytes plays a crucial role in Mn neuroinflammation. Mn activates microglia that enhance astrocyte activation through an NF-κB-dependent mechanism. A conditioned medium from primary microglial cells treated with Mn (100 µM, for 24 h) exacerbates inflammatory response in astrocytes, whereas a conditioned medium from Mn-exposed microglial treated with Bay 11–7082, an NF-κB inhibitor, blocks microglial-induced astrocyte inflammation [131]. Mn-induced NF-κB activation potentiates the inflammatory response in astrocytes, whereas IKK2 depletion in astrocytes downregulated cytokine expression in mixed glial cultures treated with 100 µM Mn for 24 h [132]. The treatment of primary neurons with an Mn-conditioned medium from IKK2 knock-out astrocytes ameliorates Mn-induced neuronal cell death, suggesting that this signaling pathway is a crucial inductor of neuronal injury [132]. Similarly, astrocyte-specific IKK2 knock-out mice exposed to Mn (50 mg/kg/day) via drinking water for 30 days show a less severe inflammatory response in astrocytes and protect neurons [133]. Thus, NF-κB mediates, at least in part, the glia–glia cell crosstalk and neuronal injury.

The inhibition of NF-κB mediates protection against Mn neurotoxicity. In SK-N-MC neuroblastoma cells, quercetin pretreatment inhibited the Mn-induced NF-κB pathway (500 µM Mn for 24 h). Similarly, quercetin reverted inflammatory markers in Mn-exposed male SD rats (15 mg/kg Mn, 8 days) [134]. Melatonin, a neurohormone in the pineal gland, protects microglial BV2 exposed to 100 µM Mn for 24 h through the inhibition of NF-κB activity [135]. Similarly, Resveratrol, a sirtuin (SIRT1) activator, downregulates Mn-induced NF-κB phosphorylation and acetylation in the hippocampus of C57BL/6 mice (200 µmol/kg Mn, 6 weeks). Thus, Resveratrol alleviates Mn-induced motor deficits and neuroinflammation via the activation of SIRT1 [136].

### 3.4. Sirtuin (SIRT) Signaling Pathway

Increasing evidence suggests that Mn exposure may affect the sirtuin (silent information regulator 2 proteins or SIRT) signaling pathway. However, the mechanism is poorly understood. Sirtuins belong to a family of conserved proteins, and deacetylase and ADP-ribosyltransferase activities are involved in redox homeostasis regulation, apoptosis, inflammatory responses, and cell proliferation [137]. This conserved family of proteins has seven members (SIRT1 to 7). Sirtuins modulate antioxidant response, regulating the NRF2 signaling pathway [138,139,140]. Studies showed that sirtuins regulate inflammatory response by deacetylating NF-κB and STAT3 [141]. Thus, sirtuins emerged as neuroprotectors against neurodegenerative diseases.

Recent works showed that sirtuins play a role in Mn neurotoxicity. In vitro and in vivo models demonstrated that Mn reduces SIRT expression and activity [76,136,142,143]. Several studies show that Mn-downregulated SIRT1 mediates neuroinflammation via NF-κB signaling pathway regulation. Mn-downregulated SIRT1 leads to an increase in acetylated NF-κB and STAT3, mediating neuroinflammation in the hippocampus from Mn-exposed C57BL/6 mice [136].

SIRTs participate in Mn-induced mitochondrial dysfunction via different mechanisms. Mn (200 µM for 24 h) induces mitochondrial dysfunction and neuronal injury through SIRT1 and SIRT3 protein and gene expression downregulation in primary neuronal cultures from the striatum [142]. In male C57BL/6 mice, Mn (200 µmol/kg, 6 weeks) downregulates SIRT1, increasing peroxisome proliferator-activated receptor-gamma coactivator 1-alpha (PGC-1α) acetylation, which binds and activates dynamin-related protein 1 (DRP-1) expression. Increased DRP1 expression mediates Mn-induced mitochondrial fragmentation [144]. Studies in Mn-exposed hippocampal neuronal HT-22 cells demonstrate that Mn induces growth arrest and DNA damage-inducible protein 34 (GADD34) expression and acetylation via decreasing SIRT1 expression. GADD34 inhibits eukaryotic translation initiation factor 2α (eIF2α), leading to mitochondrial dysfunction and apoptosis [145].

Recent studies suggest that SIRT mediates Mn-induced autophagy. In PC12 cells, Mn exposure downregulated the SIRT1 gene and protein expression, leading to apoptosis through an increase in FOXO3a [76]. In vitro studies reported that Mn induced autophagy by a mechanism that involves SIRT1 in BV2 microglial cells (up to 600 µM for 24 h) [143] and SH-SH5Y cells [146]. Mn-exposed male C57BL/6 mice (200 µmol/kg, 6 weeks) demonstrated that the SIRT1-FOXO3 pathway mediates Mn-induced autophagy [143]. Altogether, these studies showed that the activation of SIRT is a promising therapeutic target against Mn neurotoxic insults.

### 3.5. JAK/STAT Signaling Pathway

The Janus kinase (JAK) and signal transducers and activators of transcription (STAT) are mediators of the inflammatory responses [147]. JAK proteins are non-covalently bound to cytokine receptors. Upon receptor activation, JAKs recruit and phosphorylate STAT proteins. Phosphorylated STAT proteins dimerize and translocate into the nucleus, regulating gene expression [147].

Mn-induced activation of the JAK2/STAT3 signaling pathway in HAPI microglial cells (500 µM Mn for 12 h) and the striatum of male SD rats (2, 5, or 10 mg/kg, 30 days), increasing cytokine production [148]. A conditioned medium of Mn-exposed HAPI cells triggers the apoptosis of PC12 cells [97]. Similarly, Mn (200 µmol/kg, 6 weeks) induces STAT3 phosphorylation in the C57BL/6 hippocampus of mice via JNK and sSIRT1 signaling [136]. Thus, the Mn activation of the JAK2/STAT3 signaling pathway results in neuronal loss and neuroinflammation. In addition to STAT3, Mn activates STAT1 and STAT2 in BV2 microglial cells (100 µM for 24 h) [149]. Moreover, Mn-downregulated SIRT1 leads to an increase in acetylated STAT3 and STAT6 [150]. These mechanisms mediate Mn-induced inflammation.

A recent study demonstrated that STAT3 and STAT6 mediate the Mn disruption of microglial polarization through interaction with PGC-1α in Kunming mice (Mn up to 200 µmol/kg, 6 weeks) [141,150].

## 4. Autophagy

Autophagy is essential in maintaining cellular homeostasis and protein stability by eliminating misfolded proteins and damaged organelles [151]. Compelling evidence demonstrates an association between the dysregulation of autophagy and the incidence and progression of neurodegenerative diseases and neurotoxicity mechanisms [152,153,154,155].

Autophagy is considered a compensatory response to Mn neurotoxicity [156,157], while the dysregulation of autophagic pathways may be associated with neurodegeneration and cell damage due to Mn toxic effects [158]. In this respect, a low-to-moderate Mn exposure (250–750 μM Mn for 24 h) resulted in lysosomal membrane permeabilization, cathepsin release, and dysregulated autophagy with an increase in LC3 and p62 protein levels, altogether leading to cell death in murine microglial BV2 cells. Moreover, Mn exposure caused the ROS-mediated expansion of the lysosomal compartment, triggering autophagic flux, as evidenced by the formation of LC3 puncta in microglial cells. In this condition, melatonin or rapamycin treatment enhances autophagy to attenuate Mn cytotoxicity by ameliorating lysosomal dysfunction [159]. In the hippocampus of mice, Mn treatment decreased the expression levels of PGC-1α and ULK1 and significantly increased the expression levels of LC3-II and p62. All these effects were attenuated by resveratrol exerting neuroprotective and immunomodulatory effects [150]. In another study in mice, Mn exposure (200 mg/L) evidenced a robust increase in the expression of Beclin-1, LC3 II, and PINK1 proteins five weeks after exposure [160].

In contrast, the autophagic flux induced by acute Mn treatment (6.25–100 μM), resulted in the degradation of Huntingtin (Htt) aggregates, indicating that the restoration of autophagy leads to a protective effect in Huntington’s disease cell models [161]. Moreover, Mn may activate autophagy to alleviate ER stress-mediated apoptosis via the protein kinase RNA-like ER kinase PERK/eIF2α/ATF4 signaling pathway in dopaminergic SH-SY5Y cells [162].

Mn also activates the NLRP3-CASP1 inflammasome pathway in the hippocampus of mice and the BV2 microglial cell line by triggering autophagy–lysosomal dysfunction through excessive Mn accumulation. After six hours of exposure, Mn increased protein levels of autophagy-related markers (LC3-II, ATG5, BECN1, SQSTM1, and CTSB), concomitant with double-membrane autophagosomes, dysfunctional lysosomes, and mitochondrial vacuoles observed under electronic microscopy [116]. Moreover, Mn induced toxicity in microglial BV2 cells via the LRRK2-RAB10-CTSB-NLRP3 pathway-mediated autophagy dysfunction and inflammasome [120].

Consequently, the effect of Mn on autophagy appears to be concentration dependent. In conditions of mild damage at lower sub-toxic Mn concentrations, autophagy is still functional, leading to the degradation of injured organelles and cell components, favoring cell survival. When Mn excessively accumulates, the formation or degradation of autolysosomes and autophagosomes is impaired. Hence, autophagic flux is inhibited, and cathepsins are released into the cytosol, triggering cell death.

As reviewed by Yan and Xu, 2020, Mn-induced autophagy is related to α-syn oligomerization. Studies of Mn-induced neurotoxicity suggested that an increase in α-syn oligomers plays a crucial role in Mn-induced neuronal injury [163,164]. The α-syn induced by Mn is soluble and has stable intermediary oligomers rather than insoluble and mature α-syn fibrils, as in PD. As a small protein, α-syn plays a vital role in mediating several physiological functions, such as synaptic plasticity, vesicle transport, dopaminergic neurotransmission, antioxidative stress, and anti-apoptotic functions. On the contrary, the overexpression or oligomerization of α-syn promotes apoptosis by compromising the cell membrane integrity [158,164]. Furthermore, intracellular α-synuclein accumulation has been shown to modify transcription factor EB (TFEB) localization, which regulates autophagosome–lysosome interaction [165]. In murine astrocytes, Mn exposure decreased TFEB levels, leading to autophagic dysfunction and the accumulation of damaged mitochondria. Activation of autophagy by rapamycin or TFEB overexpression ameliorates Mn-induced mitochondrial respiratory dysfunction, ATP depletion, and excessive reactive oxygen species (ROS) [166]. In addition, Mn could regulate the expression of α-synuclein to interfere with the dissociation of the Beclin-1/Bcl-2 complex [164]. Beclin-1 participates in the formation of the PI3KIII complex after dissociating from Bcl-2. By inducing S-nitrosation stress, Mn increases S-nitrosylated JNK and Bcl-2 levels, thus inhibiting Bcl-2/Beclin-1 complex dissociation [167]. In a recent study, it was shown that Mn-induced α-Syn overexpression was responsible for the dysregulation of the Rab26-dependent autophagy in presynaptic neurons, thereby promoting the accumulation of injured synaptic vesicles and causing toxicity and cognitive and memory deficits in male C57BL/6 mice exposed to 100 and 200 μmol/kg Mn [168].

It is known that mitochondria are one of the principal organelles targeted by Mn-induced cellular toxicity. Mn can decrease mitochondrial membrane potential (MMP), elevate ROS production, and damage ATP synthase, decreasing ATP content [169,170] and altering mitochondrial dynamics [171]. Mn also induced ER stress-mediated apoptosis in rat striatum, which is involved in Mn neurotoxicity [172], and interfered with mitochondrial biogenesis [173]. In this respect, mitophagy is an essential protective mechanism by removing damaged mitochondria and preserving a healthy mitochondrial population involving the PINK/Parkin axis [174]. Studies have shown that Mn exposure can cause mitophagy dysregulation, resulting in the accumulation of unhealthy mitochondria. As a master regulator of the autophagy–lysosome pathway, TFEB regulates mitophagy by promoting the degradation of impaired mitochondria, which is suppressed by Mn exposure [166]. In human dopaminergic SH-SY5Y cells, Mn exposure (0.25–1.0 μM for 24 h) damaged mitochondria and augmented apoptosis. Moreover, Mn exposure increased the expression levels of PINK1 and Parkin, as well as autophagy markers LC3-II/LC3I and atg-5 and decreased the p62 level. Furthermore, the knockdown of Parkin inhibits Mn-induced mitophagy and results in the downregulation of LC3 II/LC3 I and atg5 and the upregulation of p62. Also, an increased MnCl_2_-induced apoptotic cell rate and ROS production were observed, suggesting activated PINK1/Parkin-mediated mitophagy exerts a significant neuroprotective effect against Mn-induced cell death.

Liu et al. evaluated whether Trealose (Tre) could relieve Mn-induced mitochondrial dysfunction and neuronal cell damage in mice. It was shown that Tre could relieve oxidative damage and reduce Mn-induced mitochondrial dysfunction through the induction of mitophagy and autophagy pathways (Bax/Bcl-2 and PINK/Parkin) [175]. Huang and colleagues demonstrated that Mn exposure led to a loss of MMP and apoptosis of SHSY5Y cells by enhancing LC3-I to LC3-II conversion and BCL2-interacting protein 3 (BNIP3) expression. BNIP3 is a mitochondrial outer membrane protein that mediates Mn-induced mitophagy and neurotoxicity, functioning as a mitophagy receptor protein [176]. Furthermore, Mn exposure reduced the expression of the mitochondrial protein TOMM20 and promoted interaction between BNIP3 and LC3 [177].

Mn also induces mitophagy by promoting nuclear retention of FOXO3 in SH-SY5Y cells (250 uM MnCl_2_ for 2–24 h). Increased nuclear FOXO3 translocation under Mn treatment and reduced mitochondrial autophagy in FOXO3 KO cells demonstrate that Mn-induced mitophagy may at least, in part, be mediated by FOXO3 signaling [178]. FOXO3 can stimulate PINK-1/Parkin proteins in the brain and induce mitochondrial autophagy [179]. In PC12 cells exposed to Mn (0–600 uM for 24 h), mitophagy was reverted by pretreatment with Lindl. Alkaloids (DNLAs) alleviated Mn-induced neurotoxicity and improved mitochondrial function. It was shown that DNLA significantly abolished the decrease in protein levels of both PINK1 and Parkin and mitigated the expression of autophagy markers Bax and LC3-II and the accumulation of p62 caused by Mn [180]. Recently, it was found that Mn interfered with the acetylation of FOXO3 by SIRT1 in BV2 microglial cells exposed to 600 uM of MnCl_2_ for 24 h. The autophagic flux is blocked by Mn in those conditions, decreasing FOXO3 nuclear translocation and its binding to LC3-II. These effects could be antagonized by the upregulation of SIRT1, alleviating the neuroinflammation caused by Mn in vivo and in vitro [143].

## 5. Conclusions

Mn homeostasis is critical for human health, and Mn deficiency or excess can lead to neurotoxicity. Previous studies focused on manganism caused by acute or chronic occupational exposure to Mn, but with the development of human industrial activities, the content of Mn in soil [181], water [182] air [183], and crops [184] has increased. In addition to causing manganism, Mn promotes the progression of other neurodegenerative diseases. A growing number of studies have corroborated that even low doses of Mn exposure may be associated with neurodegenerative diseases. Early life Mn exposure can not only affect neurodevelopment but also increase the risk of later neurodegenerative diseases. Therefore, it is urgent to explore the molecular mechanisms of Mn-induced neurotoxicity. In this aspect, figuring out the signaling pathways regulated by Mn exposure will provide insights for predictions of health consequences and the development of potential therapeutic interventions.

Here, we summarized the major molecular signaling pathways associated with Mn-induced neurotoxicity (Figure 1). The multifunction of certain key proteins and the interaction of different signaling pathways increase the complexity of Mn impacts in the brain and sometimes bring controversy. We grouped these signaling pathways into three major categories for better understanding, including IGF signaling pathways, neuroinflammatory signaling, and autophagy signaling pathways. For example, Mn can directly bind to insulin/IGF receptors to activate the IGF signaling pathway and regulate the activity of its downstream components (such as PI3K/Akt) and cross-related Ras/MAPK signaling pathways. These pathways can be activated with low-concentration short-term Mn supplementation or exposure, recapitulating the beneficial role of Mn as a coenzyme factor, rather than the adverse effects of heavy metal exposure. Therefore, under normal physiological conditions, Mn can maintain intracellular stress and promote cellular energy metabolism by increasing glucose uptake. However, under toxicological conditions (chronic and high-dose exposure), prolonged activation of the above pathways may not overcome the cytotoxic effects of Mn and may even elicit adverse downstream events to promote inflammation and cell death. For example, Mn neurotoxicity is related to the overactivation of cCAS-STING and NFκB signaling pathways, leading to an inflammatory response, which activates NLRP3-CASP1 inflammasome, induces microglial activation, and then causes Tau aggregation, a potential risk for AD. Meanwhile, Mn exposure can inhibit SIRT1, downregulate SIRT1 by mediating neuroinflammation by regulating NFκB, and result in mitochondrial dysfunction via increasing the expression of DRP-1 and GADD34. In addition, Mn exposure can increase α-Syn expression and suppress mitochondrial recycling via inhibiting autophagy activity and counteracting PINK1 and Parkin activity, thus increasing the risk of PD. These discoveries shed new light on the etiopathology of these neurodegenerative diseases and may be translated into novel therapeutic treatments.

Despite these findings, limitations and controversies remain for further investigation, given that these signaling pathways are highly dynamic and responsive to physiological environmental changes. Different experimental models, exposure doses, or time may have different impacts on different signaling pathways and even different changes in the same signaling pathway. For example, Mn exposure activates both the MAPK and PI3K/Akt signaling pathways [63] or only one of them [40]. Mn exposure activated the PI3K/Akt signaling in rat hippocampal [61] and PC12 cells [62] but inhibited PI3K/Akt signaling in mouse cortical neurons [65]. In addition, these pathways can also intersect with each other, such as FOXO, which is simultaneously regulated by multiple upstream factors. Secondly, animal Mn exposure experiments commonly address mechanisms of Mn toxicity in specific brain areas and lack assessment of the overall brain injury and comparison of mechanisms between various brain tissues. Multiple in vitro and in vivo experiments have shown a regulatory role for insulin or IGF signaling pathways in neurodegenerative disease models, which has been translated into clinical trials with limited success [185,186,187,188]. Therefore, it is still unclear whether the targets found in the experimental models have the same effect in humans.

With the innovation of experimental techniques and methods, more and more new targets of Mn neurotoxicity have been discovered, such as SIRT1, cCAS, STAT, and BDNF. In addition, autophagy, as a key downstream degradation pathway, has been increasingly valued for its role in Mn neurotoxicity. In contrast to previous short-term high-dose Mn exposure models, recent studies have adopted low-dose long-term Mn exposures [189,190,191,192], attempting to design experimental exposure models that mimic actual human exposure and better characterize the mechanism of Mn-induced health effects in humans.

In summary, Mn not only functions as an essential cofactor for a variety of enzymes directly involved in the physiological activity of the brain but it also acts as a potent cell signaling modifier to regulate certain key signaling pathways (Figure 1), indirectly regulating the nervous system. Mn deficiency or overexposure can disrupt the homeostasis of these signaling pathways and increase potential risks for HD, AD, ALS, and PD. With further investigation of Mn in signaling transduction, novel insights into the etiopathology of those neurological diseases and the development of efficient therapeutic interventions are expected in the near future.

## Figures and Tables

**Figure 1 cells-12-02842-f001:**
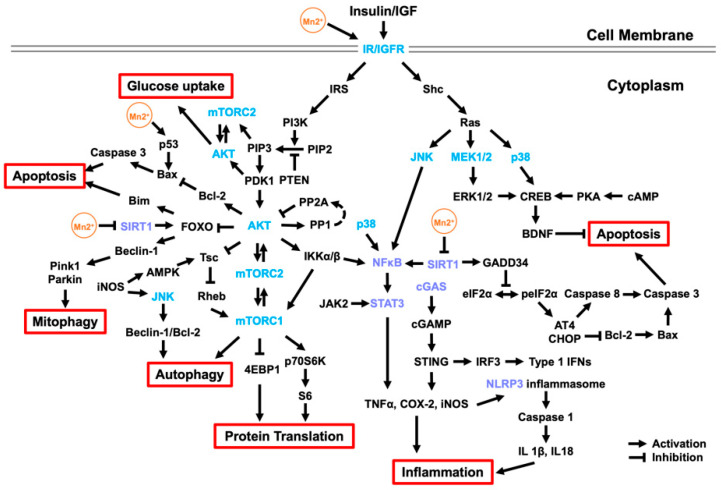
Molecular signaling pathways associated with Mn neurotoxicity. Mn, as an essential trace element, can maintain intracellular stress and promote cellular energy metabolism by increasing glucose uptake. Mn can directly bind to IR/IGFR to activate the insulin/IGF signaling pathway and regulate its downstream components, mainly PI3K/Akt, and cross-regulat the Ras/MAPK, JNK, and p38 signaling pathways (Section 2). Activated AKT can promote protein transport and activate autophagy by regulating mTOR and regulating cell apoptosis by the FOXO signaling pathway. In addition, Mn exposure can activate CREB by activating the Ras/MAPK and p38 signaling pathways, thereby increasing BDNF to inhibit apoptosis. However, prolonged activation of the above pathways may still fail to overcome the cytotoxic effects of Mn and may even elicit adverse downstream events to promote inflammation and cell apoptosis. Mn exposure can overactivate the cCAS-STING, NFκB, and JAK/STAT signaling pathways, as well as inhibit the SIRT signaling pathway, leading to an inflammatory response, which activates NLRP3-CASP1 inflammasome (Section 3). Downregulated SIRT1 can promote mitochondrial damage and apoptosis by regulating FOXO and GADD34 and further activate the NFκB and JAK/STAT signaling pathways to aggravate inflammation. Moreover, released inflammatory factors, such as iNOS, can activate autophagy through the AMPK and JNK signaling pathways. Mn also stimulates the PINK-1/Parkin protein by promoting the nuclear retention of FOXO3, inducing mitophagy (Section 4).

## Data Availability

Not applicable.

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
