# Peer review of "Signaling Pathways Involved in Manganese-Induced Neurotoxicity"

_cells, 2023, doi:10.3390/cells12242842_

Round 1

Reviewer 1 Report

Comments and Suggestions for Authors

Submitted manuscript deals that various protective strategies could be exist on neuronal injuries occurred via defect of various signaling pathways including neuroinflammation and autophagy in Mn-mediated neurotoxicity. This manuscript seems enough valuable for integration of scientific information about signaling mechanism of manganese-mediated neurotoxicity. Accordingly, this manuscript seems to be enough valuable to publish in this journal.

Minor corrections:

1.    Authors should suggest some hypothetical figures on signaling pathways related with Mn-mediated neurotoxicity.

2.    Authors should change “insulin growth factor (IGF)" into "insulin-like growth factor (IGF)" in line 86 of page 2. 

Author Response

We appreciated the reviewer's kind comments and have made corresponding revisions to address the concerns. Please find the detailed responses below and the corresponding revisions/corrections highlighted/in track changes in the re-submitted file.

Minor corrections: 

  1. Authors should suggest some hypothetical figures on signaling pathways related with Mn-mediated neurotoxicity.

       We agree and a graph was added to summarize the signaling pathways regulated by Mn. Please refer to Conclusion (Figure 1).

  1. Authors should change “insulin growth factor (IGF)" into "insulin-like growth factor (IGF)" in line 86 of page 2. 

      The text was changed accordingly.

Reviewer 2 Report

Comments and Suggestions for Authors

Manganese (Mn) is an essential trace element, but altered amounts can induce neurotoxicity. The brain is indeed, the most sensitive target organ for Mn toxicity. Mn can activate several pathways, including MAPK, Akt, ATM and mTOR both in vivo and in vitro. Since these kinases regulate transcription factors, including p53, NF-κB and FOXO, Mn can also regulate cellular function at the transcriptional level. Therefore, it is important to study the role of Mn homeostasis and related signaling pathway in Mn physiological and pathological effects. The present review summarizes the molecular signaling pathways of Mn-induced neurotoxicity, providing insights for understanding the pathophysiological roles of Mn. the paper is well-written and well organized. We believe that it may be an insightful reading for the interested researcher/clinician. My only recommendation is to summarize the presented data using a table and one or more figures.

Comments on the Quality of English Language

Manganese (Mn) is an essential trace element, but altered amounts can induce neurotoxicity. The brain is indeed, the most sensitive target organ for Mn toxicity. Mn can activate several pathways, including MAPK, Akt, ATM and mTOR both in vivo and in vitro. Since these kinases regulate transcription factors, including p53, NF-κB and FOXO, Mn can also regulate cellular function at the transcriptional level. Therefore, it is important to study the role of Mn homeostasis and related signaling pathway in Mn physiological and pathological effects. The present review summarizes the molecular signaling pathways of Mn-induced neurotoxicity, providing insights for understanding the pathophysiological roles of Mn. the paper is well-written and well organized. We believe that it may be an insightful reading for the interested researcher/clinician. My only recommendation is to summarize the presented data using a table and one or more figures.

Author Response

We appreciate the reviewer's kind comments. A graph was added to summarize the signaling pathways regulated by Mn. Please refer to Conclusion (Figure 1) of the revised manuscript for details.

Reviewer 3 Report

Comments and Suggestions for Authors

The manuscript seems to me to be of little use, lacking an analytical approach. For example, the conclusion that the authors make is too obvious and predictable:

“In summary, Mn can activate a variety of signaling pathways in the nervous system. Mn deficiency or excess can promote neurotoxicity by regulating relevant signaling pathways.”

 There is no need to list many articles to reach this conclusion.

 Modern reviews tend to be useful when accompanied by a graphical representation. The authors don't do this, which makes a bad impression. Over the past year, I found 11 reviews on the role of manganese in the brain, and the presented review can complement them too little The authors describe current issues regarding the influence of manganese on signaling pathways in cells, but do not analyze such influence and simply list these numerous signaling pathways. The authors do not suggest or describe what molecular interactions may underlie these effects. The authors do not explain how and in what form manganese penetrates into brain cells. In the text, the authors nowhere indicate chemical compounds that can penetrate the brain. In one of the latest reviews doi: 10.3390/ijms241914959, for example, there is such a description. Nowhere in the text is there any indication of the ionic form of magnesium.

Comments on the Quality of English Language

No comments

Author Response

We thank the reviewer  for taking the time to review this manuscript. We have made major revisions to address the concerns. Please find the detailed responses below and the corresponding revisions/corrections highlighted/in track changes in the re-submitted file.

The manuscript seems to me to be of little use, lacking an analytical approach. For example, the conclusion that the authors make is too obvious and predictable: “In summary, Mn can activate a variety of signaling pathways in the nervous system. Mn deficiency or excess can promote neurotoxicity by regulating relevant signaling pathways.” There is no need to list many articles to reach this conclusion.

Thank you for pointing this out, we have completely revised the summary paragraph (see line 728).

Modern reviews tend to be useful when accompanied by a graphical representation. The authors don't do this, which makes a bad impression.

We agree and a graph was added to summarize the signaling pathways regulated by Mn. Please refer to Conclusion (Figure 1).

Over the past year, I found 11 reviews on the role of manganese in the brain, and the presented review can complement them too little. The authors describe current issues regarding the influence of manganese on signaling pathways in cells, but do not analyze such influence and simply list these numerous signaling pathways. The authors do not suggest or describe what molecular interactions may underlie these effects.

We appreciate your constructive suggestions. We have revised the Conclusion, a new paragraph and a summary figure (Figure 1) were added to discuss the interactive influences of manganese on these signaling pathways related to neurodegenerative diseases (see Conclusion, line 699).

The authors do not explain how and in what form manganese penetrates into brain cells. In the text, the authors nowhere indicate chemical compounds that can penetrate the brain. In one of the latest reviews doi: 10.3390/ijms241914959, for example, there is such a description. Nowhere in the text is there any indication of the ionic form of magnesium.

Thank you for your suggestion, we added relevant content and cite this review in the Introduction (see line 50).

Round 2

Reviewer 3 Report

Comments and Suggestions for Authors

I believe that the additions made have greatly improved the manuscript and it can be published.